# Epigenetic Therapies and Biomarkers in Breast Cancer

**DOI:** 10.3390/cancers14030474

**Published:** 2022-01-18

**Authors:** Lauren Julia Brown, Joanna Achinger-Kawecka, Neil Portman, Susan Clark, Clare Stirzaker, Elgene Lim

**Affiliations:** 1School of Clinical Medicine, St. Vincent’s Campus, University of New South Wales (UNSW), Sydney, NSW 2010, Australia; Lauren.brown@svha.org.au (L.J.B.); j.achinger@garvan.org.au (J.A.-K.); n.portman@garvan.org.au (N.P.); s.clark@garvan.org.au (S.C.); c.stirzaker@garvan.org.au (C.S.); 2Garvan Institute of Medical Research, Sydney, NSW 2010, Australia

**Keywords:** breast cancer, biomarkers, epigenetics, epigenetic therapy

## Abstract

**Simple Summary:**

Epigenetic therapies are promising agents for overcoming clinical resistance to conventional treatments in breast cancer. In the assessed trials, the use of epigenetic therapies for the management of breast cancer has not translated from the pre-clinical to clinical setting. However, novel epigenetic treatments remain promising, especially in the era of personalized medicine and improved genomic evaluation. The aim of our review was to assess the published evidence for the clinical utility of epigenetic therapies and their biomarkers in breast cancer and the potential value of epigenetic biomarkers to direct clinical management.

**Abstract:**

Epigenetic therapies remain a promising, but still not widely used, approach in the management of patients with cancer. To date, the efficacy and use of epigenetic therapies has been demonstrated primarily in the management of haematological malignancies, with limited supportive data in solid malignancies. The most studied epigenetic therapies in breast cancer are those that target DNA methylation and histone modification; however, none have been approved for routine clinical use. The majority of pre-clinical and clinical studies have focused on triple negative breast cancer (TNBC) and hormone-receptor positive breast cancer. Even though the use of epigenetic therapies alone in the treatment of breast cancer has not shown significant clinical benefit, these therapies show most promise in use in combinations with other treatments. With improving technologies available to study the epigenetic landscape in cancer, novel epigenetic alterations are increasingly being identified as potential biomarkers of response to conventional and epigenetic therapies. In this review, we describe epigenetic targets and potential epigenetic biomarkers in breast cancer, with a focus on clinical trials of epigenetic therapies. We describe alterations to the epigenetic landscape in breast cancer and in treatment resistance, highlighting mechanisms and potential targets for epigenetic therapies. We provide an updated review on epigenetic therapies in the pre-clinical and clinical setting in breast cancer, with a focus on potential real-world applications. Finally, we report on the potential value of epigenetic biomarkers in diagnosis, prognosis and prediction of response to therapy, to guide and inform the clinical management of breast cancer patients.

## 1. Introduction

Epigenetics describes the molecular features that regulate gene expression, without altering the actual DNA sequence. In addition to changes in the genetic landscape, alterations in the epigenetic landscape also occur during cancer development, proliferation, treatment resistance and progression. DNA methylation, histone modifications, and chromatin remodeling are some of the key epigenetic features that are commonly altered during breast cancer progression and resistance and are, therefore, potential therapeutic targets for epigenetic therapies (Figure 1).

Whilst many potential epigenetic biomarkers show promise in predicting response to standard treatment of breast cancer, therapies that can modify the epigenetic landscape, called epigenetic therapies, have not yet made the successful translation into clinical utility. Unlike therapies that target specific genetic aberrations, epigenetic modifications are cell-type specific, making it difficult to target with great specificity and efficacy. Epigenetic therapies to date have been utilised mainly for the management of haematologic malignancies [1,2,3,4,5]. In contrast, there has been little success in solid tumours, including breast cancer, despite the wealth of preclinical evidence in support of epigenetic aberrations and the use of epigenetic therapies alone and/or in combination with other therapies [6]. Of most promise in clinical breast cancer management is the use of epigenetic therapies as primers to sensitise cancer cells to therapies to overcome de novo or acquired resistance mechanisms [7]. Thus, the role of epigenetic drugs is likely to evolve to include combinations with endocrine therapy, cytotoxic chemotherapies, targeted therapies, radiotherapy and immunotherapy. Clinical studies of epigenetics therapies that have demonstrated most promise in breast cancer include DNA methyltransferase (DNMT) inhibitors and histone deacetylase (HDAC) inhibitors. We summarise the clinical studies currently underway investigating epigenetic therapies alone or in combination with other systemic therapies for the management of the different breast cancer subtypes in Table 1. However, none of these epigenetic therapies are currently FDA or TGA approved for the management of breast cancer.

## 2. DNA Methylation and DNA Methyltransferase Inhibitors

Epigenetic regulation by DNA methylation involves the addition of a methyl group to a cytosine base in the context of a CpG dinucleotide, and is catalyzed by the family of DNMT enzymes, DNMT1 (maintenance methyltransferase) and DNMT3A and DNMT3B (DNA methyltransferases) (Figure 1). DNA methylation can play a role in transcription, remodeling of chromatin, gene imprinting, X-chromosome inactivation and suppression of repeat elements and is the most extensively characterized epigenetic modification in tumorigenesis, including in breast cancer. Overall, tumours have been shown to become demethylated globally, however normally unmethylated CpG island promoters can become methylated, which is associated with gene repression [21]. Two DNMT inhibitors, decitabine and 5-azacytadine, are currently approved for the management of haematological malignancies [3,4,5] but are not currently used in solid malignancies.

Early studies on the role of aberrant DNA methylation in breast cancer focused on increased DNA methylation (DNA hyper-methylation) of CpG islands of key cancer genes, including *ESR1* [22] and *BRCA1* [23]. Subsequent studies using DNA methylation microarrays in large breast tumour cohorts found that DNA hypermethylation at CpG islands occurs at thousands of genes in breast cancer [24], suggesting that promoter hypermethylation represents a more global trend of aberrant DNA methylation in cancer, rather than having a driver role [25]. Studies of global DNA methylation at single nucleotide resolution found widespread DNA hypomethylation in the cancer cells, primarily at partially methylated domains (PMDs) in normal breast cells [26]. Loss of DNA methylation at these domains was associated with the formation of repressive chromatin, frequently occupied by histone modifications H3K9me3 and H3K27me3, and resulting gene silencing [26]. Additionally, further studies of DNA hypomethylation at PMDs in primary breast tumours found highly variable methylation levels at these mega-base scale domains, which were further linked to other epigenetic aberrations, such as CpG island hypermethylation [27].

Recently, global DNA methylation profiling of breast tumours and normal breast tissues from the METABRIC cohort revealed both replication-dependent methylation loss in most of the genome and epigenetic instability processes modulating methylation in promoters and enhancers [28]. This highlights the important role of DNA methylation of global and regulatory elements in shaping the transcriptional aberrations in breast cancer. Several studies have reported DNA methylation at distal enhancer regions to be implicated in gene regulation, mainly by interfering with transcription factor binding to enhancer regions [29,30,31,32]. In early stage breast cancer, DNA methylation changes have been identified at the very first step of carcinogenesis and transcriptional networks associated with the estrogen receptor (ER)*, FOXA1* and *GATA3* [33], and their targets were shown to be regulated by DNA methylation at enhancers [34,35]. This is further supported by recent mechanistic work, which, through a depletion of *GATA3*, demonstrated the role of *TET2* in the maintenance of 5-hydroxymethylation at ER binding sites and in ER activity [36].

In the context of endocrine-resistant ER-positive breast cancer, altered DNA methylation profiles were observed in tamoxifen-resistant cells [37], and DNA hypermethylation was predominantly located at ER-bound enhancer regions and linked to loss of ER chromatin binding [38]. Furthermore, DNA hypermethylation and concomitant loss of ER binding at enhancers was found to be a key event associated with alterations in 3D chromatin interactions; highly dynamic ER-bound enhancer-promoter interactions have been shown to mediate expression of cancer invasion and aggressiveness genes in endocrine-resistance [39,40].

Further studies of the androgen receptor (AR) in breast cancer have demonstrated its role in chromatin binding. In endocrine-resistant breast cancer with *ESR1* aberrations, there is activation of AR leading to displacement of ER and other transcriptional co-activators from chromatin at ER-regulated cell cycle genes [41]. Whilst the role of chromatin in oncogenesis is complex, there is clearly an interplay between hormone receptors in breast cancer and the chromatin landscape. This highlights that therapeutic strategies designed to target chromatin are likely to be of promise in the management of these patients and further focus on drug development in this space is warranted [42].

In TNBC, distinct DNA methylation profiles have been observed compared to other breast cancer subtypes; altered DNA methylation was shown to be associated with the oncogenic role of DNMT1 [43]. Additionally, TNBC-specific DNA methylation signatures associated with patient’s outcome and prognosis have been identified [44]. It has also been shown that DNA methylation of repetitive DNA sequences is required to control the activation of transposable elements that can induce viral mimicry response in tumour cells [45,46]. Recent studies in TNBC found that tumour cells evade viral mimicry response by adapting their epigenetic state, where DNA hypomethylation over transposable elements is compensated by large chromatin domains of H3K27me3 that maintain transposable element repression [47,48].

Epigenetic alterations in tumour microenvironment and cancer-associated fibroblasts (CAFs) have also been reported in breast cancer. DNA methylation profiling of breast cancer tissues revealed epigenetic alterations to the tumour microenvironment, in particular DNA methylation signatures associated with immune cells that had high prognostic value in specific breast tumour subtypes [49]. Epigenetic dysregulation, including altered DNA methylation in breast CAFs have been implicated in enhanced breast cancer cell survival and therapeutic resistance [50].

The DNMT inhibitors, azacytidine and decitabine, are DNA methylation modulators being studied broadly in cancer research. Azacytidine has been shown to inhibit the proliferation of preclinical ER-positive breast cancer models, alone and in combination with doxorubicin chemotherapy [51,52]. Low dose decitabine treatment has been shown to inhibit tumour growth in ER-positive PDX models through de-methylation and re-expression of tumour suppressor genes [53,54]. Decitabine-induced DNA hypomethylation was associated with activation of ER-responsive enhancers that in turn establish new chromatin interactions with the promoters of tumour suppressor genes, leading to their activation and suppression of tumour growth [54]. Similarly, decitabine and azacytidine have also demonstrated anti-tumour activity in preclinical TNBC models [55,56]. The presence of DCK (the decitabine processing enzyme) was found to be abundant in these cells suggestive that DCK abundance may be a potential predictive biomarker which requires further evaluation. In another study, protein levels of DNMTs were also demonstrated to correlate to response [57]. There have been no published trials in the use of DNMT-inhibitors alone for the management of breast cancer to date. There are, however, clinical trials underway investigating the use of decitabine in combination with platinum chemotherapy and immunotherapy (Clinicaltrials.gov identifiers NCT03295552 and NCT02957968 respectively accessed on 1 November 2021) in TNBC and ER-positive breast cancer.

## 3. Histone Modifications and Histone Deacetylase Inhibitors

Histone modifications are an important regulatory epigenetic mechanism, with substantial evidence supporting their role in cancer development [58]. In eukaryotic cells, DNA is wrapped around nucleosomes, composed of an octamer of four core histone proteins (H3, H4, H2A and H2B). The N-terminal tails of the histone proteins can be covalently modified by post-translational modifications (PTM) which include histone methylation, acetylation, phosphorylation, ubiquitylation, and sumoylation. These modifications affect gene expression. For example, acetylation of histone lysines is generally associated with transcriptional activation. The functional consequences of methylation of histones depends on the residue and specific site that the methylation modifies; methylation of histone 3 (H3) at lysine 4 K4 is linked to transcriptional activation, while methylation of H3 at lysine K9 is associated with transcriptional repression. Histone modifications are catalysed by specific enzymes that act at the histone N-terminal tails. Histone acetylation is highly dynamic and is regulated by the opposing action of two families of enzymes, the histone acetyltransferases (HAT) and HDACs; methylation of the histone tails is regulated by histone methyltransferases (HMT) and histone demethylases (KDM) [59]. Different histone modifications impact gene expression by altering chromatin structure between an ‘open’ transcriptionally active or ‘closed’ and transcriptionally repressed state. Moreover, these histone modifications also function by recruiting specific effector proteins, such as transcriptional regulators or chromatin remodelers, which can further contribute to chromatin remodeling and altered gene expression patterns [60]. A balance between specific histone modifications maintains the epigenetic state of a normal cell and the exquisite regulation of gene expression patterns; dysregulation of these histone modifications is associated with tumour onset and progression and offers potential targets for epigenetic therapies.

There is widespread interest in the use of HDAC inhibitors, which target HDAC enzymes leading to an increase in the level of lysine acetylation, as an epigenetic therapy approach. In ER-positive breast cancer, HDAC enzymes play an important role in the transcriptional regulation at the ER and PR-mediated signalling pathway. HDAC inhibitors, such as entinostat, vorinostat (suberanilohydroxamic acid/SAHA) and dacinostat have been shown to induce growth arrest, cell cycle arrest and apoptotic cell death in preclinical ER-positive and TNBC models [61,62]. Valproic acid, a drug widely used for the management of epilepsy, is another HDAC inhibitor with anti-tumour activity demonstrated in breast cancer models in vitro [63].

HDAC inhibitors have also been shown to have combinatorial anti-tumour activity with other systemic therapies. Vorinostat, in combination with tamoxifen, demonstrated activity in tamoxifen-resistant ER-positive cell lines, potentially acting by resensitising the cells to endocrine therapy [64]. Similarly, entinostat in combination with aromatase inhibitors demonstrated greater in vivo anti-tumour activity in letrozole-resistant MCF-7 xenografts compared with either agent alone [65]. In this study, upregulation of ERα and downregulation of HER2, pHER2 and pAKT was noted in tumours that responded to entinostat, suggesting potential off-target effects of the HDAC inhibitor through the modulation of HER2 signaling rather than reversal of acquired resistance through epigenetic silencing. Additionally, entinostat has been shown to increase the expression of ERα in ER-negative models and stimulate sensitivity of these tumours to aromatase inhibitors [66]. Finally, dacinostat in combination with trastuzumab and chemotherapy has shown anti-tumour activity and a corresponding decline in HER2 and pAKT levels in HER2-amplified breast cancer cell lines [67].

The promising pre-clinical evidence for the activity of single agent HDAC inhibitors in breast cancer, however, has failed to translate into clinical studies, which have been negative to date. In a two-stage phase II trial in patients with metastatic breast cancer, the first 12 evaluable patients had no confirmed responses to vorinostat and the trial was ceased [8]. The most common toxicities included fatigue, nausea and deranged liver function tests. In contrast, HDAC inhibitors in combination with endocrine therapy has shown more clinical promise in the context of ER-positive breast cancer. A phase II trial reported a median progression free survival (PFS) of 4.28 months with entinostat plus aromatase inhibitors compared with 2.27 months with aromatase inhibitors alone (HR 0.73, 95% CI 0.50–1.07, *p =* 0.11) [15], which lead to an accelerated conditional FDA approval. Interestingly, a larger benefit was seen in the overall survival (OS) in the combination group compared to the aromatase inhibitor alone arm (28.1 months vs. 20 months; HR 0.59, 95% CI 0.36–0.97, *p =* 0.036). There were higher rates of fatigue, neutropenia and discontinuation reported in the combination group. These promising results were not replicated, however, in the subsequent phase III study [16]. The median OS was 23.4 months with combination therapy compared with 21.7 months with exemestane (HR 0.99, 95% CI 0.82–1.21; *p =* 0.94) The PFS and response rates were also similar amongst the two groups. Rates of adverse events were also higher with the combination arm including grade 3 and 4 myelosuppression. This trial was reported about 10 years following the initial phase II study and is likely to be reflective of the evolving treatment paradigms for metastatic ER-positive breast cancer during this time.

In contrast, another phase III randomised-placebo control trial with tucidinostat, a pan-HDAC inhibitor, had more promising results [14]. In this trial, patients with ER-positive breast cancer who have progressed on endocrine therapy were randomised in a 2:1 ratio to tucidinostat or placebo in combination with exemestane. The median PFS was 9.2 months with the combination versus 3.8 months with exemestane (HR 0.71, 95% CI 0.53–0.96, *p =* 0.024). The most common grade 3 or 4 adverse events were neutropenia and thrombocytopenia occurring in 51% and 27% of patients receiving the combination compared with 2% and 2% with exemestane alone respectively. Another phase II study of vorinostat in combination with tamoxifen reported a 19% objective response rate and median OS of 29 months [17]. Of those that had an objective response, all had prior exposure to aromatase inhibitors and about half had prior tamoxifen. Similar to other HDAC inhibitors, grade 3 and 4 fatigue, neutropenia and thrombocytopenia were the major toxicities noted. Exploratory biomarker analysis demonstrated an increased expression of HDAC2 and change in acetyl-4 in responders to vorinostat compared with non-responders.

The promising results for the use of HDAC inhibitors together with endocrine therapy comes with some caveats. It is important to note that at the time of these studies, the use of CDK4/6 inhibitors were not in widespread use for the management of metastatic ER-positive breast cancer. Whilst the reported haematological adverse events were high in these trials, they were reported as mostly asymptomatic and manageable. Of specific interest is the improvements in OS noted despite the relative lack of clinical response or PFS benefit, suggesting that there may be a reprogramming of the tumour rather than simply a cytotoxic mechanism at play.

There have also been multiple early phase basket studies of chemotherapy combinations with epigenetic therapies, including patients with breast cancer, with none of these trials demonstrating impressive results in breast cancer [11,12,13]. There is only one study to date with published data evaluating the combination of epigenetic therapy with cytotoxic chemotherapy which recruited breast cancer patients alone. A phase I/II study of the vorinostat, paclitaxel and bevacizumab as first line therapy in patients with metastatic breast cancer (30% TNBC, 70% ER-positive) demonstrated an overall response rate of 49% (95% CI 37–60%) with an additional 30% of patients with stable disease for longer than 24 weeks [10]. The median PFS in this study was 11.9 months and median OS was 29.4 months. The response rate for both ER-positive breast cancer and TNBC was similar to the previously published reports on the response rates of paclitaxel and bevacizumab [68], thus it is unclear of vorinostat added to the backbone systemic therapy. Similarly, vorinostat has not demonstrated much promise clinically in combination with HER2-directed therapies to date. A phase I/II trial with vorinostat and trastuzumab in patients with metastatic HER2-positive breast cancer following progression on trastuzumab reported a disappointing median PFS of 1.5 months and a median OS of 9.3 months [20].

## 4. Chromatin Remodeling, Super-Enhancers and Bromodomain and Extra-Terminal Motif Inhibitors

Chromatin remodelers alter chromatin structure and have essential roles in DNA damage repair, recombination, replication and transcriptional control. Subunits of chromatin remodelers are among the most commonly mutated genes in human cancers [69]. Of these, inactivating mutations and loss of SWI/SNF (SWItch/sucrose non-fermentable) subunits, a subfamily of ATP-dependent chromatin remodeling complexes, are the most frequent genetic alterations across many cancer types, including in breast cancer [70]. The SWI/SNF multiunit complexes remodel the chromatin structure in an ATP-dependent manner to modulate transcription and enable transcription factor binding. Several lines of evidence have demonstrated the key role of SWI/SNF in the transcriptional activation by nuclear receptors in breast cancer. The SWI/SNF component BRG1 has been shown to physically interact with ER and is required for ER-mediated transcriptional activity [71,72]. On a locus-specific level, BRG1 can bind to ER regulatory elements independently of ER [71], suggesting that the SWI/SNF complex might contribute to chromatin remodelling before ER binding. Mutations in *ARID1A*, a subunit of the SWI/SNF complex, are the most common alterations of the SWI/SNF complex in ER-positive breast cancer and are enriched in the endocrine-resistant metastatic setting [73]; loss of ARID1A promotes endocrine therapy resistance [74,75]. ARID1A determines breast luminal lineage fidelity and endocrine therapy sensitivity and influences HDAC1/BRD4 activity, intrinsic proliferative capacity and breast cancer treatment response [74,75].

Bromo- and extra-terminal domain (BET) proteins are a subfamily of bromodomain (BRD) family proteins that recognise acetylation of histones and recruit complexes such as the mediator complex and the positive transcription elongation factor β (P-TEFβ) complex to promote transcriptional initiation and transcript elongation [76]. BET-family includes the members BRD2, BRD3, BRD4 and BRDT, which are associated with transcriptional upregulation of several genes involved with cell cycle regulation, with important oncogenic potential such as *MYC*, *CCND1,* and *CCNA1*. BET family proteins contain two adjacent bromodomains (BD1 and BD2) that confer selectivity for different combinations of histone acetylation marks upon the different family members. Over the last decade, a number of small molecule inhibitors of specific BET family proteins as well as pan-BET inhibitors have been developed. These largely target BD2 but BD1-directed inhibitors and dual inhibitors are in clinical development.

BET-inhibitors are currently being evaluated in the treatment of cancer, and selectivity target tumour cells by preferentially binding to super-enhancers, noncoding regions of DNA critical for the transcription of genes that determine a cell’s identity [76]. There have been multiple pre-clinical studies in the evaluation of BET-inhibitors in TNBC with promising activity in growth inhibition in vitro and in vivo [77,78,79], in tamoxifen-resistant ER-positive breast cancer in vitro and in luminal B breast cancer mouse models [80,81]. Phase 1 studies of the BET-inhibitor mivebresib including patients with breast cancer have shown limited efficacy to date [9].

## 5. Promising Epigenetic Therapy Combinations

The combination of epigenetic therapies with immunotherapy or targeted therapies represent novel combinatorial approaches to treat cancers [42]. It is postulated that epigenetic therapies play a role in the activation of immune responses. HDAC-inhibitors and DNMT-inhibitors cause upregulation of antigens that are normally epigenetically silenced and thus can induce immune signaling. Studies of non-small cell lung cancer suggest that the use of DNMT-inhibitors leads to a series of immune-related signaling events and thus an enhanced immune response against cancer cells [82,83]. Cell line and xenograft studies in other tumour types have demonstrated increased PD-L1 expression of tumour cells when treated with decitabine, resulting in improved recruitment CD8-positive T cells and enhanced the efficacy of immunotherapy targeting PD-L1 [82,84,85]. Whilst there is evidence demonstrating efficacy of immunotherapy in the management of PDL1-positive TNBC [86,87,88], harnessing epigenetic processes may be key to improving responses in breast cancer to immune checkpoint inhibitors (ICI’s).

Preclinical models of breast, colorectal and ovarian cancer treated with low-dose azacytidine analysed using gene-set enrichment analysis (GSEA) have demonstrated alterations of a number of tumorigenesis pathways including cell cycle and mitotic pathways, SNA replication and mRNA translation and transcription [89]. The principal effect noted was the upregulation of the immune gene sets, which have been termed azacytidine immune (AIM) genes. Patients with TNBC receiving azacytidine and entinostat (Clinicaltrials.gov identifier NCT01349959;accessed on 1 November 2021) demonstrated increased expression of the AIM gene panel [89], highlighting the possible role of these agents to increase activity of immunomodulatory pathways.

In a HER2-positive breast cancer mouse model, the use of entinostat in combination with anti-HER2 therapy, anti-PD1, anti-CTLA4 or both ICI’s significantly improved survival compared with ICI or entinostat alone [90]. Treatment with ICI’s in combination with entinostat also led to a decrease in tumour burden with and without the use of anti-HER2 therapy. The mechanism of action by which entinostat and ICI’s improve survival and tumour response is postulated to be mediated through the alteration of myeloid-derived suppressor cells (MDSC). MDSC’s act to prohibit T-cell activation and infiltration and signaling functions involved in myeloid function. Thus, the use of entinostat may alter the function of MDSC allowing T-cell infiltration and availability for activation by ICI’s. In this model, using gene expression profiling and ex-vivo assays, the use of entinostat and anti-CTLA4 were associated with increased impairment of the immunosuppressive functions of MDSC’s [90]. The use of entinostat and anti-PD1 treatment downregulated the ERBB, VEGF and mTOR signalling pathways and promoted infiltration of effector CD8-positive T cells.

There are no published clinical trials in the combination of immunotherapy and epigenetic therapies to date. Entinostat combined with ICI’s in advanced solid tumours is being evaluated (Clinicaltrials.gov identifier NCT02453620; accessed on 1 November 2021), and current studies ICI’s with epigenetic therapies are summarised in Table 2.

Whilst CDK4/6 inhibitors are the current standard of care for the management of metastatic ER-positive cancer in the first-line setting, there are a lack of studies on epigenetics in the context of CDK4/6 inhibitors. These therapies have improved survival endpoints, in the first-line setting for patients with metastatic ER-positive cancer [91,92,93,94]. However, despite impressive benefits to PFS, not all patients demonstrate response to therapy and most patients relapse over time. Given the widespread use of these treatments for patients with luminal breast cancer, further understanding is required of the epigenetic context and changes that take place with CDK4/6 inhibitor therapy. As described, HDAC inhibitors play a role in HR-positive breast cancer and have shown promise in the management of patients who have developed resistance to conventional endocrine therapy [14,15,16,17]. However, these studies were performed in populations who had not been exposed to CDK4/6 inhibitor treatment. Currently, there is no consensus on the best treatment following progression on CDK4/6 inhibitors. Thus, further assessment of the use of these agents in the CDK4/6 resistant population is critical to expand therapeutic options for this population and to assess the efficacy of HDAC inhibitors amongst current therapeutic paradigms.

The use of PARP inhibitors have been shown to be of benefit to patients with *BRCA1* methylation [95]. Whilst breast cancers arising in patients with *BRCA1* or *BRCA2* mutation carriers account for less than 5% of breast cancers, if epigenetic alterations, such as *BRCA1* methylation are taken into account these make up approximately 15% of all patients and almost half of those with TNBC. PARP inhibitors are effective against cancer cells with defective DNA repair mechanisms [96].

In particular, PARP inhibitors in combination with HDAC inhibitors have demonstrated activity in pre-clinical models of TNBC [97]. TNBC cells with PTEN expression demonstrates increased sensitivity to the cell response to the combination in vitro and in vivo [98]. Olaparib in combination with BET-inhibitors has also been shown to improve the sensitivity of *BRCA* wild-type TNBC to Olaparib in vitro and in vivo [99,100], and postulated to occur as repression of BET-activity sensitizes homologous recombination-proficient tumours to PARP inhibition. A clinical trial with PARP inhibitors combined with the BET inhibitor, ZEN003694, is currently underway (Clinicaltrials.gov identifier NCT03901469; accessed on 1 November 2021).

## 6. DNA Methylation Biomarkers in Breast Cancer

Detection of breast cancer at an early stage, predicting outcome, monitoring response to therapy and detecting disease relapse, are all key to improving the outcomes for breast cancer patients. DNA methylation is one of the earliest, most stable and frequent alterations in the cancer genome [101]. Moreover, these DNA methylation alterations are large-scale, tissue and cancer-type specific and compared to the relatively low frequency of genetic mutations [101], the number of DNA methylation changes is high in cancer, translating to enhanced specificity and greater ability to identify changes associated with cancer disease and progression. Comprehensive mapping of cancer methylomes is enabling the discovery of DNA methylation signatures that offer enormous potential as molecular biomarkers to guide clinical management of breast cancer in early diagnosis and follow-up of breast cancer patients [102] (Table 3).

Molecular profiling of circulating cell-free DNA (cfDNA) or circulating-tumour DNA (ctDNA) represents an important paradigm shift in precision medicine as it provides a minimally invasive method for predictive and prognostic marker detection, as well as early and serial assessment of metastatic disease, including follow-up during remission, characterising treatment response, and monitoring minimal residual disease [103]. ctDNA reflects the same mutations, genetic and epigenetic aberrations of those of primary tumours. While cancer-specific somatic mutations are being assessed to monitor breast cancer progression in ctDNA [104,105], a major limitation is that only a few defined somatic mutations are common in breast cancer and up to 43% of patients cannot be monitored by this approach [104]. Hence the focus on DNA methylation biomarkers as novel and timely approach.

Detection of breast cancer at an early stage is of widespread interest as early diagnosis can lead to improved prognosis. Numerous studies have investigated the methylation status of breast cancer to identify methylation-based diagnostic tests for early state detection, mostly in blood-based samples, measuring methylation in a range of gene panels, and utilising different assays, such as Methylight and digital droplet MSP [106,107]. The sensitivity and specificity of these promising assays for early breast cancer has been reported in excess of 80%, comparable with that of mammography screening, and was higher in stages II and III breast cancer compared with stage I breast cancer [108,109]. The Galleri^™^ test is a multi-cancer early detection test (>50 cancer types) used to complement existing cancer screening methods [110]. It detects methylation patterns of cfDNA and can identify the tumour’s tissue of origin with high accuracy when tumour cfDNA is present [111]. It has a reported sensitivity and specificity of >95% for stages III and IV, and <50% for stages I and II breast cancer [111], highlighting limitations for early breast cancer diagnosis using this test.

A number of genes have been shown to be useful for monitoring response to therapy; for example, *BRAC1* [112], *STRATIFIN* [113], *RASSF1A* and *NEUROD1* [114,115] have been used to monitor treatment efficiency. Another cfDNA methylation signature with a sensitivity of 80% and specificity of 97% to detect breast cancer, demonstrated its utility for monitoring response during neoadjuvant therapy [116]. A similar approach has also been evaluated in patients with advanced breast cancer for prognostication and monitoring of response to systemic therapies [117,118]. Overall, these assays, while promising, have only been evaluated in small sample sizes, and now need to be validated in larger studies to be used in the routine clinical setting.

A number of epigenetic biomarkers have been reported to be associated with improved outcomes and response to endocrine therapies in ER-positive breast cancer. In a meta-analysis of 74 studies, hypermethylation of *RASSF1*, *BRCA*, *PITX2*, *CDH1*, *RARB*, *PCDH10*, *PGR*, *GSTP1*, *RASSF1* and *RARB* showed a statistically significant correlation with poor disease outcomes [119]; in another study, hypermethylation of *PTEN*, *PTGER4*, *CDK10*, *HOXC10*, *ID4*, *NAT1*, *PITX2* and *PGR* were predictive of resistance to endocrine therapy and poorer clinical outcome [120]. In contrast, hypermethylation of *ESRI and CYP1B1* have been shown to be associated with improved clinical outcomes. *PSAT1* promoter hypermethylation has been shown to predict for response to tamoxifen [121], while methylated CpG sites are associated with development of endocrine resistance [122]. DNA hypomethylation of *ESR1*-responsive enhancer elements (located within *DAXX*, *MSI2*, *NCOR2*, *RXRA*, *C8orf46*, *GATA3*, *ITPK1*, *ESR1* and *GET4* genes) is critical in endocrine-responsive ER-positive cancer, with hypermethylation of these sites associated with reduced response to endocrine therapy [38].

In TNBC, differentially methylated regions (DMRs) can be used to stratify TNBCs into methylation clusters associated with outcome [44]. Specifically, 190 CpG probes were associated with overall survival in the TNBC subset of The Cancer Genome Atlas (TCGA), highlighting the potential of DNA methylation biomarkers for disease stratification in TNBC patients. In addition, a 100-marker prognostic panel was described where high methylation was associated with an increased probability of tumour recurrence [123]. Studies of methylation biomarkers to predict chemotherapy response in TNBC are relatively limited, however. A two-gene methylation panel (*FERD3L* and *TRIP10*) was identified as a predictive biomarker for a pathological complete response following preoperative chemotherapy [124].

The focus on DNA methylation as a biomarker in cancer has led to the development of techniques to exquisitely and sensitively detect DNA methylation in clinical samples. These methodologies include targeted methods of candidate genes, gene panels or untargeted whole-genome methylation sequencing approaches. Numerous experimental methods have been used to assess and validate candidate DNA methylation biomarkers. Targeted gene approaches include bisulphite pyrosequencing [125,126], combined bisulphite restriction analysis (COBRA) [127], EpiTYPER [128], MethyLight [129], Methylation-Specific PCR (MSP) [130], Headloop-MSP [131,132,133], digital bisulphite genomic sequencing [134], digital MethyLight [134], methylation-sensitive high resolution melting and targeted multiplex bisulphite amplicon sequencing [135]. Unbiased whole-genome approaches include whole-genome bisulphite sequencing [136], and immunoprecipitation-based protocols, such as MeDIP-seq [137]. The clinical validation of these tests is of the utmost importance in order to deliver robust methylation data from fresh tissue biopsies, formalin-fixed paraffin embedded (FFPE) tissue and ctDNA to detect DNA methylated specific regions with methylation differences of 1% [106].

## 7. Discussion and Future Perspectives

Whilst not clinically utilised currently, the potential for epigenetic treatments is vast. The summarised data demonstrates epigenetic therapy in combination with endocrine therapy improves overall survival outcomes for patients. This supports the hypothesis that epigenetic treatments can potentially change the natural history of the disease. Whilst the pre-clinical studies presented demonstrate improvements in tumour control and apoptosis, from a clinical perspective this has not translated to significant improvements in progression free survival. In the reviewed investigations of epigenetic therapies in breast cancer, there is large discrepancy seen amongst the endpoints or the read outs reported in the studies. Given the limited translation of these therapies into disease control in the clinical context, there is a possibility that either the experimental readouts or the traditional endpoints for clinical trials need to be reconsidered for epigenetic therapies. Mechanistically, these treatments are targeting cancer cells in a different way to traditional drugs and aim to “reprogram” the cancer rather than cause apoptotic cell death. Certainly, the impressive overall survival data in the clinical studies presented suggests the use of epigenetic targets assists with rewiring, reprogramming or re-sensitising breast cancer to improve responses to further therapies and thus improve overall survival. Whilst apoptosis and PFS are reasonable endpoints for traditional therapies, if epigenetic treatments are assisting with “reprogramming” and, thus, slow the tumour kinetics or growth of the cancer, then a novel approach to efficacy evaluation will need to be developed.

To date, no epigenetic agent has been approved for use in the management of solid organ tumours, including breast cancer. The most interest in management of breast cancer has been in treatment of ER-positive breast cancer and the use of epigenetic therapies to overcome endocrine resistance. Given the known interplay of hormone-receptors and the chromatin landscape, chromatin targets are of great promise in this space. At present, the clinical studies to assess the efficacy of epigenetic therapies for ER-positive breast cancer, have taken place prior to the widespread implementation of CDK4/6 inhibitors. Given that CDK4/6 inhibitors are recommended in the first line for patients, further investigation of epigenetic therapies in the current clinical context is required. HDAC inhibitors seem to be of most utility in patients with ER-positive disease who have developed resistance to conventional endocrine therapy. Therefore, the place of epigenetic therapies, in particular, may be in patients with resistance to combination endocrine therapy and CDK4/6 inhibitors. Further studies in this space are warranted moving forward to understand where epigenetic therapies sit in the current therapy landscape. Furthermore, whilst the fundamental mechanisms of the drugs are known, there are other ubiquitous actions of these drugs which are not fully understood and there may be other mechanisms at play. A deeper understanding of the effect of these drugs on individual cells, tumours, tumour microenvironment, immune system and the potential “off-target” actions are required to utilise these treatments to their full potential in the clinical setting.

Whilst pre-clinical data for solid organ tumours including breast cancer has been promising, this has not been reflected in clinical settings. Importantly, epigenetic changes occurring in cell lines or PDX models may not be reflective of what occurs in humans, particularly as many preclinical models lack an intact immune system. Other possible mechanisms of failure of these drugs in the clinical setting include intra-tumoural heterogeneity and cancer cell plasticity [138]. Thus, cancer cell-intrinsic factors such as genetic, epigenetic and proteomic changes can lead to varied responses between patients. Additionally, the plasticity of breast cancer cells and their ability to rapidly adapt through genetic and epigenetic changes is also likely to explain some of the failed clinical applications of epigenetic drugs. Further understanding of cell plasticity and heterogeneity in cancer processes is required for the applications of epigenetic therapies in the clinical landscape.

The inclusion of translational and biomarker endpoints is important in the assessment of the future direction for in-human use of epigenetic therapies in breast cancer patients. Given the heterogeneity of breast cancer even within subtypes, the identification of predictive biomarkers is crucial for successful application of epigenetic drug therapies in the clinical setting. The inclusion of biomarker endpoints in the design of future clinical trials and a re-evaluation of experimental and clinical trial endpoints may be paramount to moving epigenetic therapies forward in breast cancer and to predict which patients with breast cancer may benefit from epigenetic therapies and which patients may not. To date, the most promising biomarkers appear to be in predicting treatment responses and changes whilst on epigenetic treatment. In the described studies, a reduction in HER2 expression and pAKT levels was demonstrated through multiple pre-clinical studies to be associated with responses to HDAC inhibitors. Additionally, expression of DCK, DNMT levels, histone hyperacetylation, DNA methylation and genomic assays are promising biomarkers to predict response to epigenetic treatments.

As our understanding of the genomic subtypes and treatment implications for breast cancer advance, so too will the potential applications for epigenetic therapies increase. Recently published data on multiomic-histopathological analysis of breast cancers examines the relationships between gene expression, mutation status and the clinical subtyping of patients with breast cancer [139]. These models have the potential to improve treatment stratification for patients and may also identify those that may benefit from epigenetic therapies in the future.

The clinical and preclinical results on epigenetic therapies reported to date, highlight the promise and enormous potential of epigenetic therapeutic strategies in the clinic. There are many challenges moving forward in understanding the value of combining epigenetic treatment regimens with current therapeutic strategies, addressing treatment resistance, and in understanding the potential pleiotropic effects of epigenetic-based therapies.

## Figures and Tables

**Figure 1 cancers-14-00474-f001:**
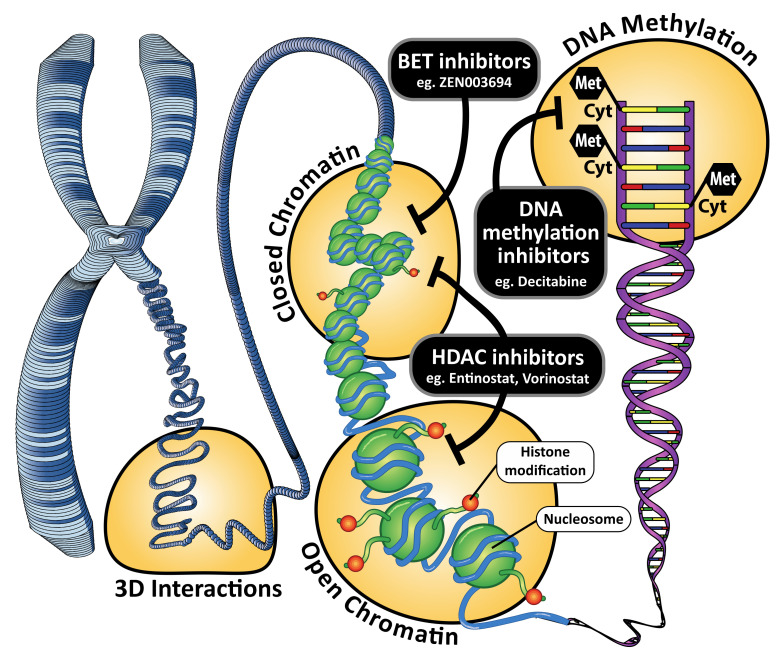
Sites of action of epigenetic therapies in breast cancer.

**Table 1 cancers-14-00474-t001:** Completed Trials of Epigenetic therapies in Breast Cancer (Adapted with permission) [6].

Therapy	Phase	Patient Population	Breast Cancer Patients (*n*)	ORR (%)	OS (months)	PFS (months)	AE’s ≥ Grade 3–4	Biomarkers	Ref
Epigenetic Therapies Alone									
*Vorinostat*	II	Advancedbreast cancer	14	0	NR	NR	Diarrhoea (7%) Nausea (7%) Dehydration (7%)	NA	[8]
*Mivebresib*	I	Solid organ tumours	8/72	0	NR	1.8	Thrombocytopenia (35%) Anaemia (6%)	NA	[9]
Epigenetic Therapies + Cytotoxic Agents									
Breast									
*Vorinostat* + paclitaxel + bevacizumab	I/II	Advancedbreast cancer	44	55	29.4	11.9	Neutropenia (27%) Anaemia (6%) Thrombocytopenia (2%) Diarrhoea (6%) Fatigue (19%)	HSP90 lysine 69-acetylation Acetyltubulin Acetylhistone H3 Acetylhistone H4	[10]
All Comer Trials									
Hydralazine + *magnesium valproate* + chemotherapy	II	Solid organ tumours	3/17	24	NR	NR	Thrombocytopenia (47%) Anaemia (41%) Neutropenia (23%) Hypoalbuminaemia (23%) Infection (23%)	Histone Deacetylation DNA demethylation	[11]
*Decitabine* + Carboplatin	I	Solid organ tumours	5/35	3%	NR	NR	Neutropenia (46%) Leucopenia (43%) Anaemia (6%) Thrombocytopenia (3%) Fatigue (9%) Mucositis (3%)	DNA methylation (dose dependent decrease) MAGE1 CpG Demethylatio	[12]
*Vorinostat* + Doxorubicin	I	Solid organ tumours	5/32	8%	NR	NR	Neutropenia (25%) Thrombocytopenia (12%) Mucositis (3%)	Histone H3 H4 acetylation HDAC2 expression	[13]
Epigenetic Therapies + Endocrine Therapy									
Breast									
Exemestane +/− *tucidinostat*	III (2:1)	Advanced ER+ breast cancer	365	18% vs. 9% (*p* = 0.025)	NR	7.4 vs. 3.8 (*p* = 0.033)	Neutropenia (51% vs. 3%) Thrombocytopenia (27% vs. 3%) Leucopenia (19% vs. 3%)	NR	[14]
Exemestane +/− *entinostat*	II	Advanced ER+ breast cancer	135	NA	28.1 vs. 19.8 m (HR 0.59; *p =* 0.036)	4.3 vs. 2.3	Fatigue (13%) Neutropenia (15%) Thrombocytopenia Higher rates of treatment discontinuation in the *entinostat* group (11% vs. 2%)	Protein lysine hyperacetylation associated with prolonged PFS in *entinostat* arm	[15]
Exemestane +/− *entinostat*	III (1:1)	Advanced ER+ breast cancer	608	5.8% vs. 5.6%	23.4 vs. 21.7 (HR 0.99; *p =* 0.94)	3.3 vs. 3.1 (HR 0.87; *p =* 0.030)	Neutropenia (20%) Hypophosphatemia (14%) Anaemia (8%) Leukopenia (6%) Fatigue (4%) Diarrhoea (4%) Thrombocytopenia (3%)	Higher increase in lysine acetylation in PMBCs in the *entinostat* arm	[16]
*Vorinostat* + tamoxifen	II	Advanced ER+ breast cancer	43	ORR 19%	29	10.3	Thrombocytopenia 9% Neutropenia 16% Fatigue 16% VTE 7%	HDAC2 expression Histone hyperaceytlation	[17]
Epigenetic Therapies + Targeted therapy									
All Comers									
*5-azacytidine* + erlotinib	I	Solid organ tumours	1/30	7%	7.5	2	Neutropenia (27%) Neuropathy (3%) Anaemia (3%) Infection (7%)	NA	[18]
*Vorinostat* + sirolimus	I	Solid organ tumours	1/70	3%	10.3	2	Thrombocytopenia (31%) Neutropenia (8%) Anaemia (7%) Fatigue (3%)	NA	[19]
Breast									
*SAHA* + trastuzumab	I/II	HER2-positive and negative metastatic breast cancer	15	7%	9.3	1.5	Thrombocytopenia 6%	NA	[20]

OS: Overall Survival. PFS: Progression Free Survival. ORR: Overall Response Rate. NR: Not reached. NA: Not Applicable. ER: Estrogen receptor. HR: Hazard Ratio. PMBCs: Peripheral Blood Mononuclear cells. VTE: Venous thromboembolism. HER2: Human Epidermal Growth Factor Receptor 2.

**Table 2 cancers-14-00474-t002:** Summary of Epigenetic therapy trials in-progress for Breast Cancer.

Epigenetic Target	Breast Cancer Subtype	Phase of Trial	Interventions	Status	Clinical Trials Reference
*DNMT*	Advanced HR+, HER2– Progressed on AI.	2	Fulvestrant + *Azacitadine*	Terminated	NCT02374099
	Locally advanced, resectable HER2–	2	Pembrolizumab + *Decitabine* followed by neoadjuvant chemotherapy	Recruiting	NCT02957968
	Advanced HER2–	1b	Paclitaxel + *Decitabine*	Unknown	NCT03282825
	Advanced TNBC	2	Carboplatin *+ Decitabine*	Recruiting	NCT03295552
	Advanced, any subtype	1b/2	Nab-paclitaxel *+ Azacitadine*	Completed	NCT00748553
*DNMT + HDAC*	Advanced HER2–	2	*Azacitadine* + *Entinostat*	Active, not recruiting	NCT01349959
*HDAC*	Advanced HR+	2	Exemestane +/− *Entinostat*	Active, not recruiting	NCT02115282
	Advanced HR+	3	Exemestane +/− *Entinostat*	Active, not recruiting	NCT03538171
	Advanced HR+	2	Exemestane +/− *Entinostat*	Active, not recruiting	NCT03291886
	Advanced HR+, HER2–	1b/2	Atezolizumab + *Entinostat* vs. Fulvestrant	Recruiting	NCT03280563
	Advanced HR+	1	Nivolumab + Ipilimumab + *Entinostat*	Active, not recruiting	NCT02453620
	Advanced HR+, PD1 > 10%	2	Tamoxifen + Pembrolizumab +/− *Vorinostat*	Active, not recruiting	NCT04190056
	Advanced HER2–and Stage I-III HER2–, with residual disease following neoadjuvant chemotherapy	1	Capecitabine + *Entinostat*	Recruiting	NCT03473639
	Advanced HER2+	1	Lapatinib + Trastuzumab + *Entinostat*	Completed	NCT01434303
	Early stage TNBC	2	Neoadjuvant Anastrozole + *Entinostat*	Terminated	NCT01234532
	Advanced TNBC	1	Ribociclib + *Belinostat*	Recruiting	NCT04315233
	Advanced TNBC	1/2	Cisplatin + Nivolumab + *Romidepsin*	Suspended	NCT02393794
*BET*	Advanced HR+	1	Fulvestrant *+ Molibresib/GSK525762*	Active, not recruiting	NCT02964507
	Advanced HR+	1	Fulvestrant or Exemestane *+ Alobresib*	Completed	NCT02392611
	Advanced TNBC, *BRCA* wildtype	2	Talazoparib + *ZEN003694*	Active, not recruiting	NCT03901469

HR: Hormone receptor. TNBC: Triple negative breast cancer. AI: Aromatase inhibitor. Phase 1 trials are breast cancer specific unless otherwise specified. Epigenetic Therapies are represented in italics.

**Table 3 cancers-14-00474-t003:** DNA Methylation Biomarkers in Breast Cancer.

Epigenetic Biomarkers	Genes with Poor Outcomes	Genes with Improved Outcomes	Methods of Assessment
Prognostic Biomarkers	Hypermethylation of *BRCA*, *PITX2*, *CDH1*, *RARB*, *PCDH10*, *PGR*, *GSTP1*, *RASSF1*, *PTEN*, *PTGER4*, *CDK10*, *HOXC10*, *ID4*, *NAT1*	Hypermethylation of *ESRI* and *CYP1B1*	Methylight Digital Droplet MSP Bisulphite pyrosequencing Combined bisulphite restriction analysis (COBRA) EpiTYPER MethyLight Methylation-Specific PCR Headloop-MSP Digital bisulphite genomic sequencing Methylation-sensitive high resolution melting Targeted multiplex bisulphite amplicon sequencing. Whole-genome bisulphite sequencing MeDIP-seq
Predictive Biomarkers	Hypermethylation of *DAXX*, *MSI2*, *NCOR2*, *RXRA*, *C8orf46*, *GATA3*, *ITPK1*, *ESR1* and *GET4* genes	Hypermethylation of *PSAT1* promoter hypermethylation, *FERD3L*, *TRIP10* *BRCA1* Hypomethylation of *ESR1*-responsive enhancer elements	

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
