# Peer review of "Epigenetic Therapies and Biomarkers in Breast Cancer"

_cancers, 2022, doi:10.3390/cancers14030474_

Round 1

Reviewer 1 Report

In the present Review, Lauren J. Brown and colleagues analyzed the state of art of epigenetic therapies in breast cancer, both in pre-clinical and clinical settings.

The Review discussed in detail the recent and most pertinent literature data. I have only a couple of comments to add more perspectives in this manuscript.

In general, I think Reviews are also a nice tool to share with people in the field an idea on how the things should move forward and what we miss to put these strategies in place. For this reason, I would add in the conclusion section a comment on how multi-omics data recently published can help in defining the potential subclassification of BC, possibly highlighting the possibility that for some of these tumors a dose-escalation strategy might be possible, and in this context epigenetic therapies could represent a less toxic way to reach a satisfactory clinical outcome. Also, I think that in this context, a specific mention should be done concerning the role of hormone receptors in shaping the chromatin landscape, including androgen receptor, as pointed out this year in Nature Medicine. Also, I'd like to include a paragraph on possible mechanisms of failure of these epidrugs in BC. For example, the hidden tumor heterogeneity hampering the efficacy of the epidrugs and the cell plasticity.

I would suggest moving the following sentence “Given the increasing use of CDK4/6 inhibitors in the met-astatic setting as first line treatment, further assessment of the use of these agents in the CDK4/6 resistant population is critical if HDAC inhibitors are to develop a clinical indication” in the paragraph 5 (promising epigenetic therapy combination) before the paragraph about PARPi and HDACi combination. Probably I would also expand a little to explain the importance of the CDK4/6 inhibitors in BC and the lack of studies on epigenetics in a contest of CDK4/6 blockade.

I would consider discussing about PARPi in a single place throughout the text, so I suggest moving this paragraph “BRCA1 methylation has also been identified as a predictive biomarker to inform which patients might benefit from PARP inhibitors. Whilst breast cancers arising in patients with BRCA1 or BRCA2 mutation carriers account for less than 5% of breast cancers, if epigenetic changes, such as BRCA1 methylation are taken into account these make up approximately 15% of all patients and almost half of those with TNBC. PARP inhibitors are effective against cancer cells with defective DNA repair mechanisms. Their use in xenograft models with confirmed BRCA1 methylation as a predictor of response, offers the potential to expand the applications of these drugs in the clinical setting.” in the upper paragraph 5, just for the sake of clarity, and to not being redundant.

Minor: revise some typos, e.g line 11 page 1, Line 110 page 4, and I am not sure about the meaning of the sentence in  line 27 page 27 “ In this review, we describe the current state of play of epigenetic targts and potential epigenetic biomarkers in breast cancer, with a focus on clinical trials with this class of epigenetic therapies” Could you rephrase?

Reviewer 2 Report

Submitted work represents a comprehensive overview of the role of epigenetics in breast cancer pathophysiology and the current utility of drugs targeting epigenetic modification to treat breast cancer. Authors discuss molecular mechanisms and the role of different epigenetic modifications (DNA methylation, histone modification) in regulation of cancer growth and progression. Moreover, authors discuss the approaches that targets these modifications and their use as a single therapy approaches or in a combination with targeted therapy or immunotherapy. The review is well written, properly referenced and provide useful high level overview. From my point of view, it would be interesting to add paragraphs on drugs targeting epigenetic modifiers not only acting on cancer cells but also acting on immune cells or other cells of tumor microenvironment (similar to  methylation in CAFs).
